# Isolation and molecular characterization of *Toxoplasma gondii* from placental tissues of pregnant women who received toxoplasmosis treatment during an outbreak in southern Brazil

**Camila E. Minuzzi**[1]*, **Luiza Pires Portella**[1], **Patricia Bräunig**[1], **Luis Antonio Sangioni**[1], **Aline Ludwig**[1], **Luciane Silva Ramos**[2], **Liliane Pacheco**[2], **Camila Ribeiro Silva**[3], **Flávia Caselli Pacheco**[3], **Ivone Andreatta Menegolla**[3], **Lourdes Bonfleur Farinha**[4], **Priscila Pauli Kist**[5], **Regina Mitsuka Breganó**[5], **Beatriz de Souza Lima Nino**[5], **Felippe Danyel Cardoso Martins**[5], **Thais Cabral Monica**[5], **Fernanda Pinto Ferreira**[5], **Isadora Britto**[5], **Ariana Signori**[5], **Kerlei Cristina Medici**[5], **Roberta Lemos Freire**[5], **João Luis Garcia**[5], **Italmar Teodorico Navarro**[5], **Cledison Marcio Difante**[6], **Fernanda Silveira Flores Vogel**[1]

**1** Laboratório de Doenças Parasitárias (Ladopar), Departamento de Medicina Veterinária Preventiva, Universidade Federal de Santa Maria (UFSM), Santa Maria, RS, Brazil, **2** Hospital Universitário de Santa Maria, Santa Maria, RS, Brasil, **3** CIEVS/DAT/CEVS/ Secretaria da Saúde do Estado do Rio Grande do Sul, Porto Alegre, RS, Brasil, **4** Vigilância Epidemiológica 4a Regional de Saúde/ Secretaria da Saúde, Santa Maria, RS, Brasil, **5** Departamento de Medicina Veterinária Preventiva, Universidade Estadual de Londrina, Londrina, Brazil, **6** Superintendência de Vigilância em Saúde, Secretaria Municipal de Saúde, Santa Maria, RS, Brasil

* camila.minuzzi03@gmail.com

## Abstract

*Toxoplasma gondii* is a protozoan that has great genetic diversity and is prevalent worldwide. In 2018, an outbreak of toxoplasmosis occurred in Santa Maria, Brazil, which was considered the largest outbreak ever described in the world. This paper describes the isolation and molecular characterization of *Toxoplasma gondii* from the placenta of two pregnant women with acute toxoplasmosis who had live births and were receiving treatment for toxoplasmosis during the outbreak. For this, placental tissue samples from two patients underwent isolation by mice bioassay, conventional PCR and genotyping using PCR-RFLP with twelve markers. Both samples were positive in isolation in mice. The isolate was lethal to mice, suggesting high virulence. In addition, the samples were positive in conventional PCR and isolates submitted to PCR-RFLP genotyping presented an atypical genotype, which had never been described before. This research contributes to the elucidation of this great outbreak in Brazil.

## Introduction

*Toxoplasma gondii* is a tissue cyst-forming protozoan capable of infecting warm-blooded animals, including humans, and is prevalent in most parts of the world [1]. It is one of the most

**Data Availability Statement:** All relevant data are within the paper and its Supporting Information files.

**Funding:** This study was financed in part by the Coordenação de Aperfeiçoamento de Pessoal de Nível Superior - Brasil (CAPES) - Finance Code 001 to CEM, PhD Fellow.

**Competing interests:** The authors have declared that no competing interests exist.

studied coccidians due to its importance in animal and human health [2,3], as well as its suitability as a model in molecular studies [1].

Although *T. gondii* is the only species of the genus *Toxoplasma* [1,4], there are various genotypes [1]. The first genotyping studies of *T. gondii* led to the description of a clonal population structure with three main lines, designated as type I, II, and III [5, 6]. Currently, there are many known genotypes that do not belong to these three clonal lineages and are called atypical. They are generally considered more virulent [7]. They are formed by sexual reproduction between gametes of different genotypes which occur in the intestine of felids [1]. In Brazil, these atypical genotypes have been widely described [8]. There are studies showing the prevalence of *T. gondii* in animals and humans [9, 10, 11], and some studies have performed the isolation and genetic characterization from cases of congenital toxoplasmosis [12, 13].

*T. gondii* infection is generally asymptomatic in humans. However, it is potentially serious when acquired during pregnancy in immunocompetent individuals, as it carries the risk of fetal transmission [14]. When congenital toxoplasmosis occurs, the protozoan can cause lesions in the fetus that range from subclinical to neurological lesions, and even fetal death or miscarriage [15, 16]. The clinical manifestation varies according to the stage of pregnancy, infection time [17], and genotype [16]. The latter makes congenital toxoplasmosis more serious in Brazil, due to infection with more virulent genotypes [18].

In 2018, an outbreak of toxoplasmosis occurred in Santa Maria, Rio Grande do Sul, with 809 confirmed cases. Of these, 114 were pregnant women who had 3 fetal deaths, 10 abortions, and 22 live births with congenital toxoplasmosis [19]. The objective of this study was to describe the isolation and molecular characterization of *T. gondii* from the placenta of two pregnant women with acute toxoplasmosis who delivered alive children and were receiving treatment for toxoplasmosis.

## Materials and methods

### Samples and clinical history

The placental tissue samples from two patients (patient 1 and patient 2) who delivered their babies at the University Hospital of Santa Maria during the toxoplasmosis outbreak in 2018, were referred to the Laboratory of Parasitic Diseases of the Federal University of Santa Maria (UFSM) for diagnostic purposes. Part of the tissue was intended for protozoan isolation, and another part for molecular tests.

According to their clinical history, both patients were positive for acute toxoplasmosis through the detection of anti-*T. gondii* IgM in Enzyme-linked Immunosorbent Assay (ELISA). The diagnosis of the two pregnant women occurred in the final trimester of gestation. Both patients received treatment and had alive children. The treatment protocol included a combination of Sulfadiazine, Pyrimethamine and Folinic Acid (SPAF). Patient 1 started receiving treatment from 35 weeks of gestation, while patient 2 received treatment from the 36th week. Both patients received treatment for four weeks and thereafter gave birth.

### Isolation through bioassay in mice

The placental tissues were subjected to peptic digestion individually, according to the technique described by Dubey, 1998 [20]. For digestion 50 g of placental tissue were used for peptic digestion. The digested material was resuspended in 5 mL of saline, and immediately after digestion, the mice were inoculated with 1 mL of the peptic digestion solution intraperitoneally. For each sample to be tested, four Swiss female mice were used, maintaining the fifth as a negative control. The animals were obtained from the Central Bioterium of the UFSM.

Mice were monitored daily for possible clinical signs of acute toxoplasmosis. When disease led to death, samples were collected from brain, heart, lung, and intraperitoneal fluid from all mice. The tissue was subjected to molecular analysis. Intraperitoneal fluid was also analyzed under a microscope with 40× magnification.

All procedures were approved by the Committee of Ethics in the Use of Animals of the Federal University of Santa Maria, under the protocol 7150250419.

### DNA extraction

DNA extraction was performed from placental tissue samples from both patients, and from mouse tissues using Wizard Genomics DNA Purification kit (Promega), following the manufacturer's instructions. In all cases, 20mg of tissues were used for DNA extraction.

### Polymerase Chain Reaction (PCR)

The PCR amplification was performed with specific primers TOX4 (CGCTGCAGGGAGGAAG ACGAAAGTTG) and TOX5 (CGCTGCAGACACAGTGCATCTGGATT) which amplified a 529 bp fragment from the *T. gondii* genome. The PCR was performed as described by Homan et al. 2000 [21]. As a positive control, tachyzoite DNA from the RH strain was used, and DNA-ase-free water was used as a negative control. A molecular marker of 100 bp (Brand—Ludwig Biotec) was used as the molecular standard size. Amplified products were visualized in the UV transilluminator after 1.5% agarose gel was stained with SYBR Safe DNA gel stain (Invitrogen).

### Analysis of restriction fragment length polymorphism (RFLP)

The genotypic characterization was performed from mouse tissues that were positive for the TOX gene (529 bp) using twelve markers (SAG 1, 5' SAG2, 3' SAG2, Alt SAG2, SAG3, BTUB, GRA6, C22-8, C29-2, L358, PK1, APICO), according to the technique described by Su et al. 2010 [22]. To do so, the extracted DNA was amplified by nested-PCR (n-PCR) technique followed by PCR-RFLP analysis. DNA target sequences were first amplified by multiplex PCR, using external primers of all markers, followed by nested-PCR using internal primers for each marker. DNA samples from standard strains, RH, ME49 and VEG were used as controls for genotypes I, II, and III, respectively.

The polymorphism of each locus was analyzed by standard RFLP bands which was used to distinguish each strain type. For this, nested-PCR products were digested with appropriate restriction enzymes for each marker, according to Su et al. 2010 [22]. The controls were also digested using the same restriction enzymes. The negative control consisted of DNAase-free water. The results obtained were compared and classified according to the genotypes present in ToxoDB (http://toxodb.org/toxo/).

## Results

*T. gondii* was isolated from the placental tissues of two patients. Within two weeks the mice presented signs indicative which acute toxoplasmosis such as apathy, bristly hair, photophobia, ascites, and death (Table 1). In addition, it was possible to identify a large amount of tachyzoites in the intraperitoneal fluid collected from the animals.

Table 1. Mouse bioassay.

| Patients | Number of inoculated mice | Number of positive mice in the bioassay | Mice life days |
|---|---|---|---|
| 1 | 4 | 4 | 11–13 |
| 2 | 4 | 4 | 12–15 |

**Table 2. Genotypic characterization of T. gondii isolates obtained from two patients during the Santa Maria toxoplasmosis outbreak compared to three other isolates [9,28].**

| Isolado | Markers | | | | | | | | | | | |
|---|---|---|---|---|---|---|---|---|---|---|---|---|
| | Sag1 | 5'Sag2 | 3'Sag2 | Sag3 | Gra6 | BtuB | C22-8 | C29-2 | L358 | PK1 | Alt.SAG2 | Apico |
| [a] P. 1 | I | I | I | I | III | III | II | III | III | I | I | I |
| [a] P. 2 | I | I | I | I | III | III | II | III | III | I | I | I |
| [b][28] | I | I | I | I | III | III | II | III | III | I | I | III |
| [c]BrI | I | I | I | III | II | I | u-1 | I | I | I | I | I |
| [c] Br II | I | I | I | III | III | III | I | III | I | II | II | III |
| [c] Br III | I | III | III | III | III | III | II | III | III | III | III | III |
| [c] Br IV | I | III | III | III | III | III | II | I | III | III | III | III |

[a] Outbreak patients isolates

[b] Isolated recently described in Rio Grande do Sul

[c] Common isolates in Brazil

As expected the samples of placental tissue as well as tissue samples from mice (brain, heart, and lung) submitted to conventional PCR showed an amplified product of 529 base pairs, confirming the presence of *T. gondii* DNA in the placenta of the evaluated patients, and in the bioassay mice.

In the genotypic characterization by the RFLP technique, the DNA analysis of *T. gondii* amplified from the tissues of mice submitted to the bioassay presented an atypical genotype, not yet described in ToxoDB. This result compared to other genotypes in Table 2.

## Discussion

Samples from animals have been widely used for isolation and genetic characterization of *T. gondii* [8]. However, in humans, this diagnosis is restricted [23], which makes it difficult to clarify the virulence of strains that infect humans and their genetic identity. In the present study conducted during the toxoplasmosis outbreak in Santa Maria, *T. gondii* was isolated from placental tissues of two patients with acute toxoplasmosis who received specific treatment in the third gestation trimester. This result is interesting since the success of *T. gondii* isolation is lower in cases of pregnant women receiving treatment [16, 24, 25]. The isolation of the protozoan species in the two patients in this study suggests that in both the cases the treatment protocol established did not prevent the protozoa from reaching the placenta, or that the congenital infection occurred even before the start of treatment.

In addition to confirming the presence of *T. gondii* in the placenta, the isolation in mice allows the virulence evaluation of genotypes present in the samples [26], since virulent strains usually cause acute infection with clinical signs in mice [3]. Signs which are characteristic of acute toxoplasmosis such as ascites, bristly hair, and photophobia were seen in all mice inoculated with placental samples from the patients in this study. In addition, the mice died within a maximum of 15 days, suggesting that the genotype present in the samples was quite virulent, although the amounts of inoculated tachyzoites can also interfere, since it was not estimated.

The genotype found in these samples was characterized as atypical, and is related to more severe forms of toxoplasmosis [27]. Atypical genotypes are not uncommon in Brazil, where the genetic diversity of *T. gondii* is large [8], but the genotype present in the samples of this research had not yet been described in ToxoDB. Recently, in Southern Brazil, Vielmo et al., 2019 [28], also described an atypical genotype very similar to that found in the current study, capable of causing a chicken outbreak on a small rural property, suggesting that these two

closely related genotypes are virulent to humans and animals. Although very similar to each other, both genotypes differ from the Brazilian clonal lineages, BrI, BrII, BrIII and BrIV [9], as shown in Table 2.

In addition, it should be considered that the evaluated patients were diagnosed in the last gestational trimester and started receiving treatment after 30 weeks of gestation. This fact reaffirms the importance of diagnosis pregnant women through serology is essential for fast and efficient treatment to reduce cases of congenital toxoplasmosis [29, 30].

This study was funded by the Higher Education Personnel Improvement Coordination.

## Conclusion

It was possible, by isolation and genotyping, to identify a new atypical *T. gondii* genotype, never described before, and with high virulence characteristics. This research contributes to elucidate the outbreak of toxoplasmosis in Santa Maria, Brazil.

## Supporting information

**S1 File. Patient Bioassay 1, Patient Bioassay 2 and Mouse images from the bioassay showing some clinical signs.**
(DOCX)

## Author Contributions

**Conceptualization:** Camila E. Minuzzi, Luiza Pires Portella, Patricia Bräunig, Luis Antonio Sangioni, Luciane Silva Ramos, Liliane Pacheco, Camila Ribeiro Silva, Flávia Caselli Pacheco, Ivone Andreatta Menegolla, Lourdes Bonfleur Farinha, Priscila Pauli Kist, Regina Mitsuka Breganó, Roberta Lemos Freire, João Luis Garcia, Italmar Teodorico Navarro, Fernanda Silveira Flores Vogel.

**Data curation:** Camila E. Minuzzi, Luiza Pires Portella, Luciane Silva Ramos, Liliane Pacheco, Camila Ribeiro Silva, Flávia Caselli Pacheco, Ivone Andreatta Menegolla, Lourdes Bonfleur Farinha, Regina Mitsuka Breganó, Beatriz de Souza Lima Nino, Felippe Danyel Cardoso Martins, Kerlei Cristina Medici, Roberta Lemos Freire, João Luis Garcia, Italmar Teodorico Navarro, Cledison Marcio Difante, Fernanda Silveira Flores Vogel.

**Investigation:** Camila E. Minuzzi, Luiza Pires Portella.

**Methodology:** Camila E. Minuzzi, Luiza Pires Portella, Patricia Bräunig, Aline Ludwig, Ivone Andreatta Menegolla, Priscila Pauli Kist, Regina Mitsuka Breganó, Beatriz de Souza Lima Nino, Felippe Danyel Cardoso Martins, Thais Cabral Monica, Fernanda Pinto Ferreira, Isadora Britto, Ariana Signori, Kerlei Cristina Medici, Fernanda Silveira Flores Vogel.

**Project administration:** Camila E. Minuzzi, Fernanda Silveira Flores Vogel.

**Supervision:** Luiza Pires Portella, Luis Antonio Sangioni, Regina Mitsuka Breganó, Roberta Lemos Freire, João Luis Garcia, Italmar Teodorico Navarro, Fernanda Silveira Flores Vogel.

**Writing – original draft:** Camila E. Minuzzi, Luiza Pires Portella, Patricia Bräunig, Aline Ludwig, Fernanda Silveira Flores Vogel.

**Writing – review & editing:** Patricia Bräunig, Aline Ludwig, Fernanda Silveira Flores Vogel.

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
