## [Decision Letter · Decision Letter 0]

24 Oct 2019

PONE-D-19-26140

Isolation and molecular characterization of Toxoplasma gondii from placental tissues of pregnant women who received toxoplasmosis treatment during an outbreak in southern Brazil

PLOS ONE

Dear Dr Minuzzi,

Thank you for submitting your manuscript to PLOS ONE. After careful consideration, we feel that it has merit but does not fully meet PLOS ONE’s publication criteria as it currently stands. Therefore, we invite you to submit a revised version of the manuscript that addresses the points raised during the review process.

Your manuscript was reviewed by 3 experts in the field. Novel methods or findings are not a requirement for publication in Plos One, but scientific rigor, technical merit, and full transparency are essential to meet the standard of the journal. Reviewers 1 and 2 commented on the need to include more detail on the methodology, study cohort, and data (details are included in their comments). These requests should be met before the manuscript can be considered for publication. 

We would appreciate receiving your revised manuscript by Dec 08 2019 11:59PM. To enhance the reproducibility of your results, we recommend that if applicable you deposit your laboratory protocols in protocols.io, where a protocol can be assigned its own identifier (DOI) such that it can be cited independently in the future. For instructions see: http://journals.plos.org/plosone/s/submission-guidelines#loc-laboratory-protocols

We look forward to receiving your revised manuscript.

Kind regards,

Leticia Reyes

Academic Editor

PLOS ONE

**Journal Requirements:**

3. As part of your revision, please complete and submit a copy of the ARRIVE Guidelines checklist, a document that aims to improve experimental reporting and reproducibility of animal studies for purposes of post-publication data analysis and reproducibility: https://www.nc3rs.org.uk/arrive-guidelines. Please include your completed checklist as a Supporting Information file. Note that if your paper is accepted for publication, this checklist will be published as part of your article.

**Comments to the Author**

1. Is the manuscript technically sound, and do the data support the conclusions?

Reviewer #1: Yes

Reviewer #2: Yes

Reviewer #3: Partly

2. Has the statistical analysis been performed appropriately and rigorously? 

Reviewer #1: N/A

Reviewer #2: N/A

Reviewer #3: N/A

3. Have the authors made all data underlying the findings in their manuscript fully available?

Reviewer #1: No

Reviewer #2: No

Reviewer #3: Yes

4. Is the manuscript presented in an intelligible fashion and written in standard English?

Reviewer #1: No

Reviewer #2: No

Reviewer #3: No

5. Review Comments to the Author

Reviewer #1: This manuscript reported isolation and genetic characterization of Toxoplasma gondii strains isolated from congenital toxoplasmosis from an outbreak occurred in Santa Maria, Brazil in 2018, from which over 800 people were exposed to the parasite. Placenta tissues from two congenital toxoplasmosis were bioassayed in mice and the parasites were isolated from both cases. Genotyping by multilocus PCR-RFLP markers revealed a new genotype. The genotype seems to be highly virulent to mice.

The results of this study is of interest to molecular epidemiology of T. gondii, particularly for this largest known outbreak of toxoplasmosis in history.

The manuscript is overall well written, however, editing is needed to correct grammar errors.

There is a recent paper reported isolation of T. gondii in chicken from the same state (near Porto Alegre, Rio Grande Do Sul) (Andréia Vielmo et al., Outbreak of toxoplasmosis in a flock of domestic chickens (Gallus gallus domesticus) and guinea fowl (Numida meleagris). Parasitology Research, 2019, 118, 991–997). The genotype is almost identical to the one identified in this manuscript. The authors should look into the paper and discuss the results in that context.

Specific Comments:

Line 33. It should be mice not rats.

Lines 91-93. How much placenta tissue was used for pepsin treatment, 1 gram, 2 grams, or more? What was the volume the digested tissues were finally resuspended and then 1 ml was inoculated to each mouse?

Use a table to summarize the results from mouse bioassay, including how many mice were inoculated, how many were positive with T. gondii and the days mice are euthanized.

Table 1. Was genotyping performed using DNA samples extracted directly from placentas? If so, list these samples in the table.

Lines 162-164. It can be argued that the congenital infection has already occurred before the drug treatment, not necessary of the failure of the drug treatment.

Lines 165-171. Given that the amount of parasites inoculated to mice is not known in this study, the result of high virulence could partially due to high inoculation does.

Lines 173-175. The statement “It is necessary to consider a combination of alleles and formation of an atypical genotype may have contributed to an increase in virulence (Dardé, 2008)” is way beyond the scope of this study. Remove this sentence.

Reviewer #2: The authors report isolation and molecular characterization of a strain isolated from placentas of 2 pregnant women infected during a large outbreak observed in 2018 in Rio Grande do Sul. The 2 isolates were identical with PCR-RFLP markers, suggesting a common source. This genotype seems different from the multiple ones already described in Brazil.

Although this report may be interesting as few isolates from human cases of toxoplasmosis are characterized in Brazil, it would have been more interesting if clinical data (mothers and babies) were made available to discuss the possible role of this genotype in the severity (or not) of clinical toxoplasmosis.

Apparently, the description of the large outbreak of Santa Maria is not yet published, and the paper presented here is just the contribution of the authors to the exploration of this outbreak. However, a few other information regarding isolates would be interesting. Are the 2 isolates described here the only ones available from this outbreak?

- L. 55-56: “In Brazil …. isolation and genetic characterization of T. gondii from human tissues, especially from pregnant women, are scarce.”: yes, but they do exist. Could the authors add Brazilian references, at least for congenital cases? (Carneiro et al., 2013 J. Clin. Microbiol. 51, 901–907; Ferreira et al., 2011 Exp. Parasitol. 129, 190–195; Silva et al., 2014 PloS One 9, e90237). The authors could discuss about the comparison with other strains isolated from congenital cases in Brazil.

- L.84: “the diagnosis of the two pregnant women occurred in the final trimester of gestation.”: does that mean that infection occurred in the last trimester or just that the diagnostic was made at this stage?

- L. 145: “Detection of protozoal DNA in mouse tissues also indicates the feasibility of T. gondii infection in placental tissues.”: do you mean the presence of Toxoplasma infection in placenta, not the feasibility?

- Genotype: the authors described the genotype obtained as a new genotype for Brazil, according to ToxoDB. It would be interesting to show a dendrogram indicating the place of this genotype among other genotypes described in Brazil. Or at least, add to table 1, the PCR-RFLP profiles of the main Brazilian genotypes (Br I to IV) and other genotypes found in Rio Grande do Sul state in animals.

- L.160: “This result is uncommon, since only a few researchers have been able to carry out the T. gondii isolation from pregnant women tissues,”: this is not true and reports of isolation of T. gondii form placenta are very common (see for example, Ajzenberg et al. JID 2002; Robert-Gangneux et al. 2010)

Reviewer #3: There are articles with the molecular characterization of approximately 50 T. gondii strains , it seems to me very little to publish an article with the characterization of 2 strains.

There are more recent methodologies for molecular characterization such as microsatellites and NGS.

Isolation of T. gondii strains from biological samples such as amniotic fluid placentas and newborn blood by inoculation in mice is a routine and ancient procedure used by European reference laboratories.

6. PLOS authors have the option to publish the peer review history of their article (what does this mean?). If published, this will include your full peer review and any attached files.

Reviewer #1: Yes: Chunlei Su

Reviewer #2: No

Reviewer #3: No

---

## [Author Response · Author response to Decision Letter 0]

13 Nov 2019

Reviewer #1: This manuscript reported isolation and genetic characterization of Toxoplasma gondii strains isolated from congenital toxoplasmosis from an outbreak occurred in Santa Maria, Brazil in 2018, from which over 800 people were exposed to the parasite. Placenta tissues from two congenital toxoplasmosis were bioassayed in mice and the parasites were isolated from both cases. Genotyping by multilocus PCR-RFLP markers revealed a new genotype. The genotype seems to be highly virulent to mice.

The results of this study is of interest to molecular epidemiology of T. gondii, particularly for this largest known outbreak of toxoplasmosis in history.

The manuscript is overall well written, however, editing is needed to correct grammar errors.

There is a recent paper reported isolation of T. gondii in chicken from the same state (near Porto Alegre, Rio Grande Do Sul) (Andréia Vielmo et al., Outbreak of toxoplasmosis in a flock of domestic chickens (Gallus gallus domesticus) and guinea fowl (Numida meleagris). Parasitology Research, 2019, 118, 991–997). The genotype is almost identical to the one identified in this manuscript. The authors should look into the paper and discuss the results in that context.

Reply: 

Thank you for reminding us of this study. It will be added in our discussion.

Specific Comments:

Line 33. It should be mice not rats.

Reply: 

Reviewed

Lines 91-93. How much placenta tissue was used for pepsin treatment, 1 gram, 2 grams, or more? What was the volume the digested tissues were finally resuspended and then 1 ml was inoculated to each mouse?

Reply: 

50g of placental tissue were used for peptic digestion.

The digested material was resuspended in 5mL Saline.

Use a table to summarize the results from mouse bioassay, including how many mice were inoculated, how many were positive with T. gondii and the days mice are euthanized.

Reply: 

Table 1. Mouse bioassay.

Patients Number of mice inoculated Number of positive mice in the bioassay Mice life days

P. 1 4 4 11 - 13

P. 2 4 4 12 - 15

Table 1. Was genotyping performed using DNA samples extracted directly from placentas? If so, list these samples in the table.

Reply: 

The table referring to genotyping will now be table 2 as another table has been included in the manuscript. Genotyping was performed only from the tissues of positive mice in the bioassay.

Lines 162-164. It can be argued that the congenital infection has already occurred before the drug treatment, not necessary of the failure of the drug treatment.

Reply: 

Will be added to the text. Line 165.

Lines 165-171. Given that the amount of parasites inoculated to mice is not known in this study, the result of high virulence could partially due to high inoculation does.

Reply: 

Will be added to the text. Line 173.

Lines 173-175. The statement “It is necessary to consider a combination of alleles and formation of an atypical genotype may have contributed to an increase in virulence (Dardé, 2008)” is way beyond the scope of this study. Remove this sentence.

Reply: 

The sentence will be removed.

Reviewer #2: The authors report isolation and molecular characterization of a strain isolated from placentas of 2 pregnant women infected during a large outbreak observed in 2018 in Rio Grande do Sul. The 2 isolates were identical with PCR-RFLP markers, suggesting a common source. This genotype seems different from the multiple ones already described in Brazil.

Although this report may be interesting as few isolates from human cases of toxoplasmosis are characterized in Brazil, it would have been more interesting if clinical data (mothers and babies) were made available to discuss the possible role of this genotype in the severity (or not) of clinical toxoplasmosis.

Apparently, the description of the large outbreak of Santa Maria is not yet published, and the paper presented here is just the contribution of the authors to the exploration of this outbreak. However, a few other information regarding isolates would be interesting. Are the 2 isolates described here the only ones available from this outbreak?

Reply: 

This study is really just a contribution to elucidating the causes of the great outbreak that occurred in Santa Maria. We agree that clinical data from patients would make the research more interesting, but not the central objective of the research. Thus we limit ourselves to isolation in mice and genotyping.

The 2 isolates described in this study come from the samples to which we had access. Thus we selected samples sent for diagnosis in our laboratory of patients who had received treatment for toxoplasmosis and who had children born alive.

- L. 55-56: “In Brazil …. isolation and genetic characterization of T. gondii from human tissues, especially from pregnant women, are scarce.”: yes, but they do exist. Could the authors add Brazilian references, at least for congenital cases? (Carneiro et al., 2013 J. Clin. Microbiol. 51, 901–907; Ferreira et al., 2011 Exp. Parasitol. 129, 190–195; Silva et al., 2014 PloS One 9, e90237). The authors could discuss about the comparison with other strains isolated from congenital cases in Brazil.

Reply: 

Thank you for the indication of the studies. They will be added to the manuscript.

- L.84: “the diagnosis of the two pregnant women occurred in the final trimester of gestation.”: does that mean that infection occurred in the last trimester or just that the diagnostic was made at this stage?

Reply:

We cannot say when the infection occurred. Only that the diagnosis was made at this stage.

- L. 145: “Detection of protozoal DNA in mouse tissues also indicates the feasibility of T. gondii infection in placental tissues.”: do you mean the presence of Toxoplasma infection in placenta, not the feasibility?

Reply:

Exactly. We refer to the presence of Toxoplasma in the placenta. This sentence will be corrected in the manuscript.

- Genotype: the authors described the genotype obtained as a new genotype for Brazil, according to ToxoDB. It would be interesting to show a dendrogram indicating the place of this genotype among other genotypes described in Brazil. Or at least, add to table 1, the PCR-RFLP profiles of the main Brazilian genotypes (Br I to IV) and other genotypes found in Rio Grande do Sul state in animals.

Reply:

A table with this data will be added to the discussion of the manuscript.

Table 3: Genotypic characterization of T. gondii isolates obtained from two patients during the Santa Maria toxoplasmosis outbreak compared to three other isolates [9,28].

Isolado Markers

 Sag1 5’Sag2 3’Sag2 Sag3 Gra6 BtuB C22-8 C29-2 L358 PK1 Alt.SAG2 Apico

a P. 1 I I I I III III II III III I I I

a P. 2 I I I I III III II III III I I I

b Vielmo et al. I I I I III III II III III I I III

c BrI I I I III II I u-1 I I I I I

c Br IV I III III III III III II I III III III III

This is the Table 3 legend. 

a Outbreak patients isolates

b Isolated recently described in Rio Grande do Sul

c Common isolates in Brazil

- L.160: “This result is uncommon, since only a few researchers have been able to carry out the T. gondii isolation from pregnant women tissues,”: this is not true and reports of isolation of T. gondii form placenta are very common (see for example, Ajzenberg et al. JID 2002; Robert-Gangneux et al. 2010)

Reply: 

This sentence will be corrected in the manuscript.

Reviewer #3: There are articles with the molecular characterization of approximately 50 T. gondii strains , it seems to me very little to publish an article with the characterization of 2 strains.

There are more recent methodologies for molecular characterization such as microsatellites and NGS.

Isolation of T. gondii strains from biological samples such as amniotic fluid placentas and newborn blood by inoculation in mice is a routine and ancient procedure used by European reference laboratories.

Reply: 

We understand that if the number of isolates surveyed were larger, the study would be more interesting. However, we are limited to the number of samples that have arrived at our diagnostic laboratory. Unfortunately we could not include more isolates in our search.

Regarding the technique used, we chose RFLP because it is a reliable technique widely used by researchers in the field. Besides being the technique available in our laboratory.

We know our research has limitations, but we are dealing with samples from what appears to be the largest outbreak of toxoplasmosis ever described in the world. This makes us think that although we have had access to few isolates, the results of isolation and genotyping are interesting and cannot be ruled out. With this we intend to make a small contribution to the elucidation of the factors that contributed to the occurrence of this outbreak.

---

## [Decision Letter · Decision Letter 1]

13 Dec 2019

PONE-D-19-26140R1

Isolation and molecular characterization of Toxoplasma gondii from placental tissues of pregnant women who received toxoplasmosis treatment during an outbreak in southern Brazil

PLOS ONE

Dear Dr Minuzzi,

Thank you for submitting your manuscript to PLOS ONE. After careful consideration, we feel that it has merit but does not fully meet PLOS ONE’s publication criteria as it currently stands. Therefore, we invite you to submit a revised version of the manuscript that addresses the points raised during the review process.

We would appreciate receiving your revised manuscript by Jan 27 2020 11:59PM. To enhance the reproducibility of your results, we recommend that if applicable you deposit your laboratory protocols in protocols.io, where a protocol can be assigned its own identifier (DOI) such that it can be cited independently in the future. For instructions see: http://journals.plos.org/plosone/s/submission-guidelines#loc-laboratory-protocols

We look forward to receiving your revised manuscript.

Kind regards,

Leticia Reyes

Academic Editor

PLOS ONE

Reviewers' comments:

Reviewer's Responses to Questions

**Comments to the Author**

1. If the authors have adequately addressed your comments raised in a previous round of review and you feel that this manuscript is now acceptable for publication, you may indicate that here to bypass the “Comments to the Author” section, enter your conflict of interest statement in the “Confidential to Editor” section, and submit your "Accept" recommendation.

Reviewer #1: All comments have been addressed

Reviewer #2: (No Response)

2. Is the manuscript technically sound, and do the data support the conclusions?

Reviewer #1: Yes

Reviewer #2: Partly

3. Has the statistical analysis been performed appropriately and rigorously? 

Reviewer #1: N/A

Reviewer #2: N/A

4. Have the authors made all data underlying the findings in their manuscript fully available?

Reviewer #1: Yes

Reviewer #2: Yes

5. Is the manuscript presented in an intelligible fashion and written in standard English?

Reviewer #1: (No Response)

Reviewer #2: No

6. Review Comments to the Author

Reviewer #1: Line 186. Suggest revise to “ … capable of causing a chicken outbreak on a small rural property, suggesting these two closely related genotypes are virulent to humans and animals.”

Reviewer #2: The authors replied to most of the reviewers’comments. The limitations of the study remain to be highlighted, notably the absence of clinical data regarding these 2 cases.

Abstract: “In addition, the samples were positive in conventional PCR and, when submitted to PCR-RFLP genotyping, presented an atypical genotype,…”: this sentence is ambiguous, since PCR-RFLP was not performed directly on placenta DNA extract but on the strain isolated in mice.

L. 104: “In all cases, 20mg of tissues were used for DNA extraction”. For placenta, was it 20mg of tissues or 20 mg of the pellet obtained after peptic digestion?

L. 191: “both genotypes differ from the Brazilian clonal lineages, BrI, BrII, BrIII and BrIV [9], as shown in table 2”: only Br1 and BrIV are presented in table 2. BrII and III should be added to table 2. Comparaison with other strains isolated from congenital cases in Brazil was not discussed.

Ref 19: not accessible, the link does not work.

The text contains many errors in English that need to be corrected.

7. PLOS authors have the option to publish the peer review history of their article (what does this mean?). If published, this will include your full peer review and any attached files.

Reviewer #1: No

Reviewer #2: No

---

## [Author Response · Author response to Decision Letter 1]

9 Jan 2020

Reply to reviewers

Reviewer #1: 

Line 186. Suggest revise to “ … capable of causing a chicken outbreak on a small rural property, suggesting these two closely related genotypes are virulent to humans and animals.”

Reply: 

Thank you for the consideration. The sentence has been changed in the manuscript.

Reviewer #2: 

Abstract: “In addition, the samples were positive in conventional PCR and, when submitted to PCR-RFLP genotyping, presented an atypical genotype,…”: this sentence is ambiguous, since PCR-RFLP was not performed directly on placenta DNA extract but on the strain isolated in mice.

Reply: 

Thank you for the consideration. The sentence has been amended in the abstract to improve understanding.

L. 104: “In all cases, 20mg of tissues were used for DNA extraction”. For placenta, was it 20mg of tissues or 20 mg of the pellet obtained after peptic digestion?

Reply:

For extraction of placental DNA, 20mg were used directly from the placental tissue.

L. 191: “both genotypes differ from the Brazilian clonal lineages, BrI, BrII, BrIII and BrIV [9], as shown in table 2”: only Br1 and BrIV are presented in table 2. BrII and III should be added to table 2. 

Reply:

BrII and III strains were added to the table.

Ref 19: not accessible, the link does not work.

Reply: The link has been changed.

The text contains many errors in English that need to be corrected.

Reply:

The text has been subjected to a new English revision. We hope we have managed to improve the language.

---

## [Editor Report · Decision Letter 2]

14 Jan 2020

PONE-D-19-26140R2

Isolation and molecular characterization of Toxoplasma gondii from placental tissues of pregnant women who received toxoplasmosis treatment during an outbreak in southern Brazil

PLOS ONE

Dear Dr Minuzzi,

Thank you for submitting your manuscript to PLOS ONE. After careful consideration, we feel that it has merit but does not fully meet PLOS ONE’s publication criteria as it currently stands. Therefore, we invite you to submit a revised version of the manuscript that addresses the points raised during the review process.

We would appreciate receiving your revised manuscript by Feb 28 2020 11:59PM. To enhance the reproducibility of your results, we recommend that if applicable you deposit your laboratory protocols in protocols.io, where a protocol can be assigned its own identifier (DOI) such that it can be cited independently in the future. For instructions see: http://journals.plos.org/plosone/s/submission-guidelines#loc-laboratory-protocols

We look forward to receiving your revised manuscript.

Kind regards,

Leticia Reyes

Academic Editor

PLOS ONE

Additional Editor Comments (if provided):

We are pleased to conditionally accept your manuscript after the following editorial corrections have been made in the body of the manuscript:

Page 4, lines 72-73 please replace "who who delivery babies their at the University Hospital" with "who delivered their babies at the University Hospital"

Page 6, line 135 please replace "T. gondii wes isolate" with T. gondii was isolated"

Page 8, line 165 please replace "Santa Maria, was isolated T. gondii from placental tissues" with "Santa Maria, T. gondii was isolated from placental tissues"

Page 9, line 175 please replace "art characteristic" with "are characteristic"

Page 9, line 185 "the corrent" should be "the current"

---

## [Author Response · Author response to Decision Letter 2]

14 Jan 2020

Response to Reviewers

Page 4, lines 72-73 please replace "who who delivery babies their at the University Hospital" with "who delivered their babies at the University Hospital"

Page 6, line 135 please replace "T. gondii wes isolate" with T. gondii was isolated"

Page 8, line 165 please replace "Santa Maria, was isolated T. gondii from placental tissues" with "Santa Maria, T. gondii was isolated from placental tissues"

Page 9, line 175 please replace "art characteristic" with "are characteristic"

Page 9, line 185 "the corrent" should be "the current"

Response:

 Thank you very much for the considerations. All of them were made in the manuscript.

---

## [Editor Report · Decision Letter 3]

16 Jan 2020

Isolation and molecular characterization of Toxoplasma gondii from placental tissues of pregnant women who received toxoplasmosis treatment during an outbreak in southern Brazil

PONE-D-19-26140R3

Dear Dr. Minuzzi,

We are pleased to inform you that your manuscript has been judged scientifically suitable for publication and will be formally accepted for publication once it complies with all outstanding technical requirements.

With kind regards,

Leticia Reyes

Academic Editor

PLOS ONE
---

## [Editor Report · Acceptance letter]

23 Jan 2020

PONE-D-19-26140R3 

Isolation and molecular characterization of *Toxoplasma gondii* from placental tissues of pregnant women who received toxoplasmosis treatment during an outbreak in southern Brazil 

Dear Dr. Minuzzi:

I am pleased to inform you that your manuscript has been deemed suitable for publication in PLOS ONE. Congratulations! Your manuscript is now with our production department. 

With kind regards,

on behalf of

Dr. Leticia Reyes 

Academic Editor

PLOS ONE